

# Patient-reported fatigue in patients with rheumatoid arthritis who commence biologic therapy: a longitudinal study

Hege Selheim Rinke[1,2], Clara Beate Gram Gjesdal[1,3], Heidi Markussen[2,4], Jörg Assmus[5] and Gerd Karin Natvig[2]

[1] Department of Rheumatology, Haukeland University Hospital, Bergen, Norway
[2] Department of Global Public Health and Primary Care, University of Bergen, Bergen, Norway
[3] Department of Clinical Science, University of Bergen, Bergen, Norway
[4] Department of Thoracic Medicine, Haukeland University Hospital, Bergen, Norway
[5] Centre for Clinical Research, Haukeland University Hospital, Bergen, Norway

Corresponding author
Hege Selheim Rinke,
hegerinke@gmail.com

## ABSTRACT

**Aims and objectives**. To examine changes in patient-reported fatigue, over a twelve month period, in rheumatoid arthritis patients who commence biologic treatment, and to identify possible predictors for such changes.

**Background**. Fatigue is a burdensome symptom for patients with rheumatoid arthritis. Despite biologics being effective in reducing disease activity, patients still report fatigue.

**Design**. A longitudinal observational study.

**Methods**. A total of 48 patients were enrolled in the study. Fatigue was measured by the Fatigue Severity Scale. Independent samples $T$-tests were used to test gender differences, and paired samples $T$-tests were used to measure differences between repeated measures. Bivariate and multiple regression analyses were used to examine potential predictors for changes in fatigue, such as age, sex, Disease Activity Score 28, pain and physical and emotional well-being.

**Results**. Forty-seven patients completed the study. From baseline to 12-month follow-up, fatigue decreased significantly in both women and men. Analyses of predictors were performed step-wise, and the final model included sex and physical well-being. The results from this final step showed that female sex was the only significant predictor for changes in fatigue.

**Conclusion**. Patients commencing biologic therapy reported a significant reduction in fatigue. Female sex was a significant predictor of changes in fatigue.

**Relevance to clinical practice**. Despite improvements in pharmacological treatment, patients with rheumatoid arthritis still report fatigue. This is a multifaceted health problem encompassing personal and emotional factors in addition to the clinical factors directly connected to the disease.

## INTRODUCTION

Rheumatoid arthritis (RA) is an inflammatory joint disease which may cause joint damage, disability and fatigue (*Scott, Wolfe & Huizinga, 2010*). RA patients experience fatigue as

unpredictable, overwhelming and different from normal tiredness (*Feldthusen et al., 2013*). Fatigue is caused by numerous factors. This might due to immunological disorders, hormonal imbalances, decrease in oxygenation, and exogenous processes induced by drugs (*Dupond, 2010*). A conceptual model for fatigue suggests inter-relationships between and within the RA disease process, personal issues, feelings, thoughts and behaviors (*Hewlett et al., 2011a*). Fatigue in RA is under-recognized and undertreated (*Hewlett et al., 2005*). Furthermore, it is one of the most burdensome symptoms from the patient perspective (*Kirwan et al., 2007*; *Van Tuyl et al., 2016*). Over the last decades biologic agents have caused a paradigm shift in the treatment of RA, and biologics are effective in reducing disease activity, inflammation, pain and joint damage in RA (*Scott, Wolfe & Huizinga, 2010*). This reduction of disease-related components may affect the level of RA fatigue, and previous research has shown that both anti-tumor necrosis factor (anti-TNF) and non-anti-TNF biologics produce similar improvements in fatigue (*Almeida et al., 2016*). However, patient-reported consequences of disease activity may differ from the assessments made by health professionals (*Studenic et al., 2012*).

## BACKGROUND

Recommendations endorsed by the European League Against Rheumatism and the American College of Rheumatology encourage all clinical trials to report fatigue (*Aletaha et al., 2008*; *Kirwan et al., 2007*). Fatigue is a patient-reported measure and can incorporate one single item or multiple items. Furthermore, the scales can have a unidimensional or multidimensional structure. The Fatigue Severity Scale (FSS) is a disease-specific questionnaire intended to assess fatigue in multiple sclerosis and systemic lupus patients, but it is also used in RA studies (*Diniz et al., 2017*; *Gok Metin & Ozdemir, 2016*; *Hussain et al., 2015*). The FSS is better at detecting changes than generic questionnaires (*Hewlett, Dures & Almeida, 2011b*). As fatigue in RA patients is measured by various patient-reported outcome measures (*Hewlett, Dures & Almeida, 2011b*; *Pouchot et al., 2008*), and as results from these measures are difficult to compare, the review below will mainly refer to research based on the FSS.

### Effect of biologic therapy on fatigue

Only a few previous studies have examined the effect of biologic therapy on fatigue measured with a disease-specific scale. In a double-blinded study, patients with Primary Sjögren's syndrome received biologic therapy or placebo, and fatigue was measured using both the FSS and Visual Analogue Scale (VAS). After four weeks of treatment there was no significant reduction in fatigue in these patients (*Norheim et al., 2012*). On the other hand, another study has evaluated the effect of biologic therapy on work ability, fatigue and functional disability in RA patients after six months. In this study, fatigue was measured using the FSS and VAS, and the results demonstrated that biologics had a beneficial effect on fatigue in patients with RA (*Hussain et al., 2015*).

### Predictors for changes in fatigue

As stated by Ahmed and colleagues, patient-reported outcomes are important as they represent information from the patient perspective that has not been interpreted by health

personnel (*Ahmed et al., 2012*) and such measures might provide additional and different information that is relevant for both RA patients and physicians (*Gossec, Dougados & Dixon, 2015*).

Previous research has identified both patient-reported factors and more objective measures evaluated by health personnel as predictors for fatigue. In a review containing both cross-sectional and longitudinal studies, as well as several measures of fatigue, the results showed a correlation between fatigue and patient-reported pain, physical function and depression (*Nikolaus et al., 2013*). In a systematic review of cross-sectional, observational and cohort studies examining psychological factors as predictors for fatigue, there was a consistent correlation between patient-reported mood and fatigue, with low mood being associated with increased fatigue (*Matcham et al., 2015*). The most common health personnel-reported measure of disease activity in patients with RA is the Disease Activity Scale 28 (DAS28) (*Van Riel, 2014*). To our knowledge, no previous studies including both patient-reported and health personnel-reported measures have measured fatigue using the FSS. In a previous study of both patient-reported and health personnel-reported outcomes, results showed that disease activity, pain, sleep disturbance, and mental health were related to fatigue (*Thyberg, Dahlstrom & Thyberg, 2009*). A review of correlations between different disease activity measures, pain and fatigue showed that pain was the strongest factor associated with fatigue (*Madsen et al., 2016*). In these studies gender has not been considered as a possible predictor for change in fatigue. In a study of patients in remission or with low disease activity, pain was a significant predictor of fatigue (*Olsen et al., 2016*).

Regarding sociodemographic data, gender differences have been observed in previous research. Female patients reported significantly higher fatigue measured by the FSS compared with healthy controls (*Buyuktas et al., 2015*), and female participants reported more persistent fatigue after four years than men did (*Druce et al., 2015a*). In a study, *Thyberg, Dahlstrom & Thyberg (2009)* found that women reported more fatigue measured by the VAS than men. Furthermore, one study found a difference between the patient and physician assessment of global disease activity, and this difference was more pronounced in women than in men (*Lindstrom Egholm et al., 2015*).

The aims of the present study were:

- To examine changes in patient-reported fatigue in RA patients who commence biologic treatment;
- To identify possible predictors (sociodemographic as well as patient-reported and health personnel-reported variables) for changes in fatigue.

## METHODS

### Design

The study was carried out at a rheumatology specialist department in Western Norway. It was a longitudinal study comparing fatigue levels over 12 months. Patients were assessed at baseline (T0) and after 3 (T1), 6 (T2) and 12 months (T3). The study was part of an observational study to explore ultrasonographic differences in total synovitis between

seropositive and seronegative rheumatoid arthritis patients. The patients consented to participate in both studies at the same time.

Inclusion criteria were as follows and the same as in the main study: (1) male or non-pregnant, non-nursing female; (2) age between 18 and 75 years; (3) patient classified as having RA according to the 2010 American College of Rheumatology/ European League Against Rheumatism criteria (*Aletaha et al., 2010*); (4) treating rheumatologist and patient have decided that biologic treatment is needed; (5) patient has had no prior biologic treatment; and (6) patient is able and willing to give written informed consent and comply with the requirements of the study protocol. Exclusion criteria: (1) abnormal renal function (serum creatinine >142 µmol/L in female and >168 µmol/L in male, or GFR <40 mL/min/1.73 m$^2$; (2) abnormal liver function (ASAT/ALAT > 3 times normal), active or recent hepatitis, cirrhosis; (3) major co-morbidities like severe malignancies, severe diabetic mellitus, severe infections, uncontrollable hypertension, severe cardiovascular disease (New York Heart Association Functional Class 3–4) and/or severe respiratory disease; (4) leukopenia and/or thrombocytopenia; (5) inadequate birth control, pregnancy, and/or breastfeeding; (6) indications of active tuberculosis; and (7) psychiatric or mental disorders, alcohol abuse or other abuse of substances, language barriers or other factors which make adherence to the study protocol impossible.

## Data collection

During the period from October 2011 to December 2014, all eligible patients were invited to enter the study. A physical examination, including checking for co-morbidities and joint counting, was performed by a rheumatologist. A study nurse coordinated the physical examinations and blood testing, and managed the patient-reported questionnaires. When the last enrolled patient had been followed for 12 months the study was closed. The number of participants is low, because the study was a subproject under the multicenter study "Ultrasonographic differences in total synovitis between seropositive and seronegative rheumatoid arthritis patients", and the Fatigue Severity Scale was not included in all subprojects.

## Treatment

The patients in this study commenced their first biologic therapy (certolizumab, etanercept, golimumab, infliximab or rituximab) according to standard procedures and doses.

Thirty-eight patients were on stable doses of methotrexate 3 months before baseline and until visit T1. Six patients were taking leflunomide or hydroxychloroquine, and four patients had no synthetic disease-modifying antirheumatic drugs (DMARDs).

A total of 28 patients were on a stable low dose of corticosteroids the last month before inclusion and until visit T1. Patients were told to avoid analgesics for 24 h prior to the visits if possible.

## Assessments

Fatigue was measured using the FSS, which is a 9-item questionnaire rated on a scale from 1 to 7, where 1 indicates strongly disagree and 7 indicates strongly agree. The FSS contains statements on the severity of fatigue, and also the effect on a person's activities and lifestyle.

The developers of the FSS have suggested a cut-off of 4 for severe fatigue (*Krupp et al., 1989*). The FSS is used in a number of diseases and is a reliable instrument for measuring fatigue (*Valko et al., 2008*). The FSS has demonstrated good psychometric properties and is one of the few measures that are able to detect change over time (*Whitehead, 2009*). There are different Norwegian versions of the Fatigue Severity Scale questionnaire. In this study we used a questionnaire translated in 1995. As far as we know, this version of the questionnaire has not been tested for psychometric properties. Nevertheless, it has been used in research and clinical practice for several years (*Johansen et al., 2018*; *Asprusten et al., 2018*; *De Rodez Benavent et al., 2017*).

Rheumatoid Arthritis Impact of Disease (RAID) was used to measure pain and physical and emotional wellbeing (*Gossec et al., 2009*). The scale goes from 0–10. Pain is assessed from none to extreme and physical and emotional wellbeing are assessed from very good to very bad. RAID has been validated (*Heiberg et al., 2011*).

Disease activity can be measured by the Disease Activity Score 28 (DAS28). The purpose is to combine single measures such as swollen and tender joints, erythrocyte sedimentation rate (ESR) or C-reactive protein (CRP), and to include or exclude a general health assessment (*Fransen, Stucki & Van Riel, 2003*). In this study we wanted to use an objective tool for measuring disease activity. Therefore, we chose DAS28 based on three variables: tender and swollen joint counts and ESR (*Fransen, Creemers & Van Riel, 2004*). DAS28 has been validated to monitor disease activity in RA (*Van Riel, 2014*).

## Statistical analyses

Descriptive statistics were used to describe sociodemographic, patient-reported and health personnel-reported variables. The independent samples $t$-test was used to test for differences between women and men. The paired samples $t$-test was used to test for differences between measures at different points in time.

Linear mixed effect analyses were used to identify possible predictors of change in fatigue. In the analyses, we added clinical variables such as DAS28, sociodemographic variables such as sex and age, and patient-reported variables such as pain and physical and emotional wellbeing. First, a 0-model containing time only was made using simple contrasts in time domain, i.e., a comparison of each time point with baseline. In step 1, to estimate the main effect and interaction effect, each predictor was put into a model containing time. The Akaikes information criterion (AIC) was used as a criterion to decide whether the model fitted the data and also to compare the models to the 0-model by measuring the $p$-value and performing a likelihood ratio test. In step 2, significant predictors were put into a model one by one. RAID subscales were added in order of their AIC score. In step 3 only predictors contributing to the model were added.

We used a linear mixed effects model with simple contrasts including all time points, i.e., the model has a built-in-adjustment for baseline. The main effect of a predictor is the effect at baseline, while the interaction describes if and how the baseline effect changes over time.

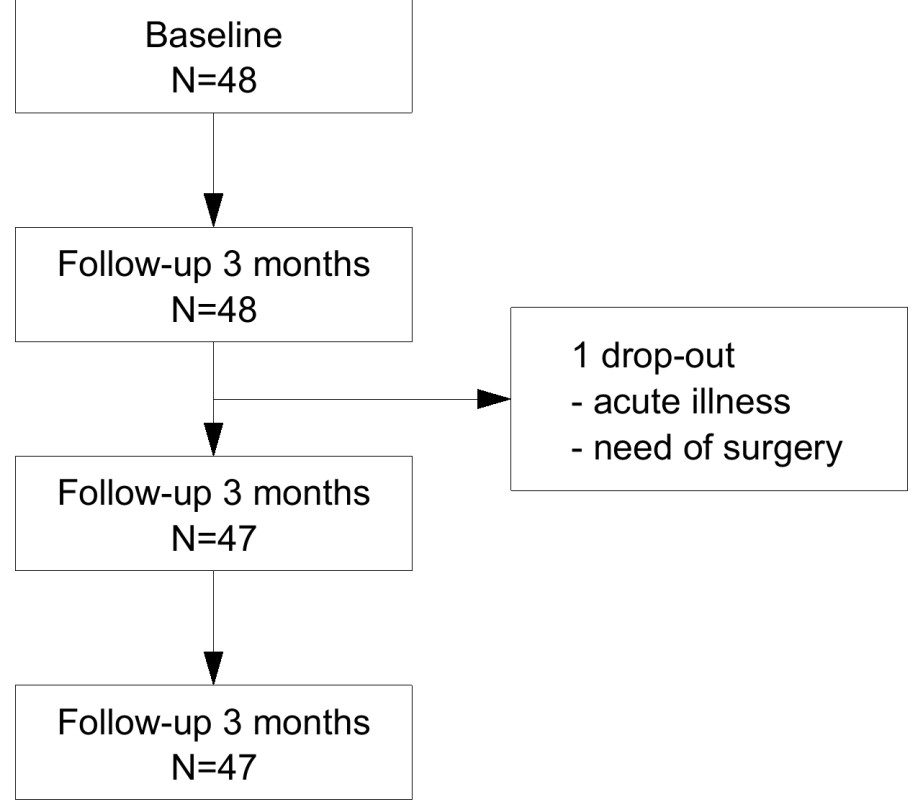

**Figure 1** **Flowchart.** Flowchart to show number of participants who completed the study.

The significance level was set to 0.05. SPSS 23 for Windows (IBM Corp., Armonk, NY) and R 3.3.0 (*R Core team, 2016*) with the package nlme 3.1 (*Pinheiro et al., 2016*) were used for the statistical analyses.

### Ethical considerations

The study was approved by the Regional Ethics committee for Medical Research (REK, 2011/490). All the patients received both oral and written information, and signed informed consent.

## RESULTS

A total of 48 patients met the inclusion criteria, and gave consent to participate in the study. One patient was excluded after 3 months because of acute illness and need of surgery. 47 patients completed the study (Fig. 1).

At baseline the patients had a median age of 55 years [range 24–73 years], and more than half of the patients (56%) were women. The mean disease duration was 5 years (SD 7.5), [range <1–40 years]. Sociodemographic and clinical baseline characteristics are shown in Table 1.

The mean fatigue measurements and their changes are shown in Table 2. At baseline we observed significantly more severe fatigue in female than in male patients. The severity of

**Table 1** **Sociodemographic characteristics such as age, sex, civil status, children living at home, occupational activity.** Clinical characteristics such as disease duration, rheumatoid factor, anti-cyclic citrullinated peptide, erythrocyte sedimentation rate, C-reactive protein, Disease Activity Score of 28 joints, Fatigue Severity Scale, Rheumatoid Arthritis Impact of Disease.

| Sociodemographic characteristics | $n = 48$ |
|---|---|
| Age, years, median (range) | 55.0 [24–73] |
| Sex, female, $n$ (%) | 27 (56%) |
| Married/living with partner, $n$ (%) | 34 (71%) |
| Children living at home, $n$ (%) | 22 (46%) |
| Working or studying (full-time or part-time), $n$ (%) | 24 (50%) |
| Working or studying patients on sick leave, $n$ (%) 10 (21%) | |
| Disability benefits (full-time or part-time), $n$ (%) | 10 (21%) |
| Retired, $n$ (%) | 4 (8%) |
| **Clinical characteristics** | |
| Disease duration, years, mean (SD) | 5.0 (7.5) |
| RF, $n$ (%) | 33 (69%) |
| Anti-CCP, $n$ (%) | 38 (79%) |
| ESR (mm/h), median (range) | 22.5 [0–75] |
| CRP (mg/L), median (range) | 9.0 [0–58] |
| Prednisolon dosage, mean (SD)[a] | 6.0 (3.2) |
| Methotrexate dosage, mean (SD)[b] | 20.0 (4.8) |
| FSS, (1–7), mean (SD)[c] | 4.4 (1.5) |
| DAS28, mean (SD) | 4.5 (1.2) |
| RAID - pain, mean (SD)[d] | 5.5 (2.1) |
| RAID - emotional well-being, mean (SD)[e] | 3.9 (2.1) |
| RAID - physical well-being, mean (SD)[d] | 4.9 (1.9) |

**Notes.**

RF, rheumatoid factor; Anti-CCP, anti-cyclic citrullinated peptide; ESR, erythrocyte sedimentation rate; CRP, C-reactive protein; DAS28, Disease Activity Score of 28 joints; FSS, Fatigue Severity Scale; RAID, Rheumatoid Arthritis Impact of Disease.

[a] $n = 28$.
[b] $n = 38$.
[c] $n = 47$.
[d] $n = 45$.
[e] $n = 44$.

fatigue decreased significantly for both women and men between baseline and visit T1 and then stabilized. This improvement was stronger for women (mean (CI) = 1.3 (0.7,1.9), $p < 0.001$) than for men, (mean (CI) = 0.6 (0.8,1.2), $p = 0.026$).

As shown in Table 3 the disease activity measured by DAS28 decreased significantly between baseline and 3 months (mean (CI) = 1.4 (1.1,1.7), $p < 0.001$) and was stable in later visits. The same development was observed for the selected RAID subscales. In the linear mixed effects model, sex and RAID emotional well-being contributed significantly to the model (Step 2 in Table 4), while change in fatigue was not explained by disease activity (Step 1 in Table 4).

Analysis of predictors (Table 4) showed a higher reduction in fatigue values at follow-up visit T2 (6 months) for women than men ($p = 0.019$). At follow-up visit T3 (12 months) there was a significant change in fatigue for females ($p = 0.015$). The changes in fatigue and

**Table 2** Mean, 95% confidence intervals and *p*-values of changes in fatigue during the study, and test of differences between women and men.

| Time point | Missing | Women FSS n | Mean (95% CI) | FSS change from T0 Mean (95% CI) | p | Men FSS n | Mean (95% CI) | FSS change from T0 Mean (95% CI) | p | FSS difference Men/women p |
|---|---|---|---|---|---|---|---|---|---|---|
| Baseline (T0) | 1 | 26 | 5.0 (4.5,5.5) | – | – | 21 | 3.6 (2.9,4.3) | – | – | 0.001[*] |
| 3 months (T1) | 0 | 27 | 3.7 (3.1,4.4) | 1.3 (0.7,1.9) | <0.001[†] | 21 | 3.0 (2.3,3.6) | 0.6 (0.8,1.2) | 0,026[†] | 0.113[*] |
| 6 months (T2) | 0 | 27 | 3.4 (2.7,4.1) | 1.6 (0.9,2.3) | <0.001[†] | 20 | 3.0 (2.3,3.7) | 0.6 (−0.3,1.5) | 0,155[†] | 0.368[*] |
| 12 months (T3) | 2 | 25 | 3.7 (2.9,4.4) | 1.4 (0.7,2.0) | <0.001[†] | 20 | 2.7 (1.9,3.5) | 0.9 (0.1,1.6) | 0,024[†] | 0.078[*] |

Notes.

FSS, Fatigue severity scale, scale 0–7: lower scores represent less fatigue; CI, Confidence interval.

[†]paired *t*-test.

[*]*t*-test.

**Table 3** Mean changes, 95% confidence intervals and *p*-values during the one-year observation study of fatigue, including both women and men.

| Measure | T1: change from T0 Mean (95% CI) | p | T2: change from T0 Mean (95% CI) | p | T3: change from T0 Mean (95 CI) | p |
|---|---|---|---|---|---|---|
| DAS28 | 1.4 (1.1,1.7) | <0.001[†] | 1.4 (1.1,1.8) | <0.001[†] | 1.6 (1.3,1.9) | <0.001[†] |
| RAID pain[a] | 2.4 (1.6,3.3) | <0.001[†] | 2.2 (2.8,0.4) | <0.001[†] | 3.2 (2.3,4.0) | <0.001[†] |
| RAID physical well-being[b] | 1.8 (1.1,2.5) | <0.001[†] | 2.0 (1.2,2.7) | <0.001[†] | 2.3 (1.5,3.1) | <0.001[†] |
| RAID emotional well-being[b] | 1.4 (0.7,2.1) | <0.001[†] | 1.7 (1.0,2.4) | <0.001[†] | 1.3 (0.3,2.2) | 0.012[†] |

Notes.

T0, before the intervention; T1, after three months; T2, after six months; T3, after twelve months; DAS28, Disease activity score 28; RAID, Rheumatoid Arthritis Impact of Disease; CI, Confidence interval.

[a]Scale 0–10, lower scores represent less pain.

[b]Scale 0–10, lower scores represent more well-being.

[†]paired *t*-test.

the RAID variables pain and physical and emotional wellbeing are explained by the gender component. The change in fatigue is explained by both female sex and physical wellbeing, but in the end female sex had stronger influence than physical wellbeing and turned out to be a significant predictor for change in fatigue ($p = 0.010$).

# DISCUSSION

## Changes in patient-reported fatigue

This study found that both female and male RA patients commencing biologic therapy reported lower levels of fatigue during treatment. Previous research has shown somewhat inconsistent results and, to our knowledge, has not examined gender differences. In a randomized clinical trial, *Norheim et al. (2012)* reported no significant effect of biologics on fatigue in Sjögren's patients. However, a post hoc analysis showed that six out of 12 patients in the group treated with biologics reported a 50% reduction in fatigue compared to one out of 13 in the placebo group, and this result was significant. Another study investigated the effect of biologics on work ability, functional disability and fatigue. The results from this observational study showed significant improvements in fatigue after six months of biologic therapy (*Hussain et al., 2015*). These inconsistencies may be explained

Rinke et al. (2019), *PeerJ*, DOI 10.7717/peerj.6771

**Table 4  Predictors of fatigue: mean changes, 95% confidence intervals and *p*-values, including both women and men.**

| Predictor | Effect type | Step 1[a] | | Step 2[b] | | Final model[c] | |
|---|---|---|---|---|---|---|---|
| | | B (95%CI) | *p*-value | B (95%CI) | *p*-value | B (95%CI) | *p*-value |
| DAS28 | Main effect | 0.06 (−0.26, 0.38) | ,706 | – | – | – | – |
| | Effect change: BL ->3 months | 0.34 (−0.09, 0.76) | ,126 | – | – | – | – |
| | Effect change: BL ->6 months | 0.13 (−0.32, 0.59) | ,569 | – | – | – | – |
| | Effect change: BL ->12 months | −0.02 (−0.46, 0.43) | ,937 | – | – | – | – |
| Age | Main effect | −0.04 (−0.08, 0) | ,081 | – | – | – | – |
| | Effect change: BL ->3 months | 0 (−0.04, 0.04) | ,856 | – | – | – | – |
| | Effect change: BL ->6 months | 0.03 (−0.02, 0.07) | ,234 | – | – | – | – |
| | Effect change: BL ->12 months | 0.02 (−0.03, 0.06) | ,450 | – | – | – | – |
| Sex | **Main effect** | **1.49 (0.54, 2.43)** | **,003** | **1.27 (0.34, 2.2)** | **,013** | **1.29 (0.36, 2.23)** | **,010** |
| | Effect change: BL ->3 months | −0.56 (−1.45, 0.34) | ,235 | 0.09 (−0.17, 0.34) | ,524 | 0.12 (−0.08, 0.32) | ,255 |
| | Effect change: BL ->6 months | −1.13 (−2.05, −0.21) | ,019 | 0.03 (−0.16, 0.22) | ,802 | −0.57 (−1.46, 0.32) | ,220 |
| | **Effect change: BL ->12 months** | **-0.58 (−1.52, 0.36)** | **,236** | **0.04 (−0.16, 0.25)** | **,687** | **-1.19 (−2.11, −0.27)** | **,015** |
| RAID physical well-being | Main effect | 0.2 (0.01, 0.39) | ,047 | −0.69 (−1.56, 0.18) | ,144 | −0.38 (−1.31, 0.55) | ,440 |
| | Effect change: BL ->3 months | 0.13 (−0.13, 0.38) | ,336 | −1.15 (−2.03, −0.28) | ,016 | 0.19 (−0.06, 0.45) | ,155 |
| | Effect change: BL ->6 months | 0.04 (−0.21, 0.28) | ,773 | −0.38 (−1.27, 0.5) | ,423 | 0.14 (−0.11, 0.4) | ,284 |
| | Effect change: BL ->12 months | −0.06 (−0.32, 0.2) | ,649 | 0.15 (−0.19, 0.49) | ,418 | 0.02 (−0.25, 0.28) | ,912 |
| RAID emotional well-being | Main effect | 0.14 (−0.03, 0.31) | ,113 | −0.15 (−0.54, 0.23) | ,464 | – | – |
| | Effect change: BL ->3 months | 0.16 (−0.08, 0.39) | ,199 | −0.12 (−0.52, 0.27) | ,561 | – | – |
| | Effect change: BL ->6 months | 0.31 (0.07, 0.56) | ,016 | 0.09 (−0.21, 0.39) | ,591 | – | – |
| | Effect change: BL ->12 months | 0.02 (−0.2, 0.23) | ,882 | 0.39 (0.09, 0.68) | ,015 | – | – |
| RAID pain | Main effect | 0.1 (−0.08, 0.29) | ,283 | 0.04 (−0.26, 0.35) | ,782 | – | – |
| | Effect change: BL ->3 months | 0.08 (−0.17, 0.32) | ,540 | 0.01 (−0.25, 0.28) | ,919 | – | – |
| | Effect change: BL ->6 months | 0.14 (−0.1, 0.38) | ,257 | 0.11 (−0.23, 0.45) | ,541 | – | – |
| | Effect change: BL ->12 months | 0.08 (−0.19, 0.35) | ,552 | 0.14 (−0.18, 0.46) | ,422 | – | – |

**Notes.**

[a]Model with one predictor (main effect and interaction).

[b]Model including sex, RAID physical well-being, RAID emotional well-being, RAID pain.

[c]Model including sex, RAID physical well-being.

BL, baseline; CI, confidence interval; DAS28, Disease Activity Score of 28 joints; RAID, Rheumatoid Arthritis Impact of Disease.

by the fact that fatigue is a patient-reported symptom with individual variations in severity and etiology and by the different diseases studied. In some patients, fatigue may persist despite biologic therapy (*Emery, 2014*) and the explanation for this may be found in the etiology of fatigue as a symptom with multiple causes, some connected to disease activity and others to personal factors (*Hewlett et al., 2011a*). When RA patients with fatigue were interviewed and encouraged to describe this problem in their own words, they described fatigue as an experience that was always present, preventing them from finding solutions to everyday problems and affecting both themselves and their social life (*Bala et al., 2016*). A broader approach covering all aspects of this health problem is needed.

## Possible predictors for changes in fatigue

In bivariate analyses, disease activity and age turned out to be insignificant predictors for change in fatigue at all follow-up visits. Pain also turned out to be an insignificant predictor for change in fatigue in both bivariate and multivariate analyses. Emotional wellbeing turned out to be a significant predictor for change in fatigue at the 6-month follow-up visit in bivariate analyses and at the 12-month follow-up visit in multivariate analyses. In bivariate analyses physical well-being was a statistically significant predictor of change in fatigue, but in multivariate analyses only the 3-month follow-up visit showed a significant change in fatigue. Female sex was a significant predictor in both bivariate and multivariate analyses. As far as we know, in previous research sex has rarely been a variable in analyses of predictors of change in fatigue, and the results of this study may be difficult to compare to other studies. However, it might be possible to compare the results of the patient and health personnel-reported outcomes in this study. In a systematic review, *Madsen et al. (2016)* found that disease activity was positively related to fatigue when pain was not considered, and that pain was the dominating factor related to fatigue. However, in these studies disease activity was measured using different components of DAS28, and the various components of DAS28 have different weightings, with some of them being more related to inflammation than others. It might therefore be difficult to compare the results of these studies (*Madsen et al., 2016*). In this study, women reported statistically significantly higher fatigue at baseline than men. During the study the mean fatigue score was higher in women at all follow-up visits. Previous work has shown that women report higher values of fatigue than men (*Rat et al., 2012*; *Thyberg, Dahlstrom & Thyberg, 2009*), and several factors, such as genetic and hormonal factors and other exposures that may be experienced differently by women and men, have been suggested as explanations for the difference between men and women in terms of RA disease impact (*Van Vollenhoven, 2009*). Pain and related **measurements** are often discussed as being **non-sex-neutral**. In a review, somatic symptom reporting in women and men has been examined. Results showed that women reported more numerous, more intense and more frequent bodily symptoms than men (*Barsky, Peekna & Borus, 2001*). Moreover, women and men may react differently to treatment. In a register-based observational study of predictors of response to biologic therapy, there was a lower remission rate among female RA patients (*Hyrich et al., 2006*).

In this present study, multivariate analyses showed that change in fatigue is explained by both female sex and physical wellbeing. Still, in the final model only female sex turned out to be a significant predictor for change in fatigue.

### Strengths and limitations of the study

The strength and importance of this study of fatigue in patients with rheumatoid arthritis is the insight and information gained on patient-reported outcomes. The data are based on the implementation of outcome measures recommended by experts (*Hewlett, Dures & Almeida, 2011b*), and guidelines from the European League Against Rheumatism and the American College of Rheumatology (*Aletaha et al., 2008*), and recommendations from the international research cooperation Outcome Measures in Rheumatoid Arthritis Clinical Trials (OMERACT) (*De Wit et al., 2013*). This study is a cohort study, without a control group. Therefore, it is difficult to determine whether biologic therapy affects fatigue or not. On the other hand, the patients in the study had tried standard treatment with synthetic DMARDs before commencing biologics. The patients' level of fatigue was followed up for twelve months, and data collection was performed four times, and this may provide valuable insight into how fatigue occurs. Furthermore, the number of participants was small, which means that the analyses have low statistical power, and they were all recruited from the same rheumatology department. However, the participants were recruited consecutively and were all in need of their first biologic treatment, and had no major co-morbidities. The patients in the study live along the west coast of Norway, but we assume the selection is not very different from the majority of RA patients living in other parts of Norway (*Brinkmann et al., 2018*; *Olsen et al., 2016*).

## CONCLUSION

Female RA patients commencing biologics report reductions in fatigue after 3 and 6 months. After 12 months there is a slight increase in the fatigue level. Male RA patients report reductions in fatigue after 3 and 12 months. When comparing sociodemographic, patient-reported and health personnel-reported variables, female sex was a significant predictor of changes in fatigue. This result is important and may indicate gender differences in the impact of RA. Further research is needed in order to understand the complexity of fatigue and to evaluate non-pharmacological treatment.

### Relevance to clinical practice

Fatigue is a burdensome symptom in RA patients, and despite improvements in the pharmacological treatment of RA, patients still report fatigue (*Van Hoogmoed et al., 2013*; *Druce et al., 2015b*; *Madsen et al., 2016*). Therefore, additional therapies are needed to combat fatigue. These therapies should take into account that fatigue is a multifaceted health problem encompassing personal and emotional factors in addition to the clinical factors directly connected to the disease.

### Funding
The authors received no funding for this work.

### Competing Interests
The authors declare there are no competing interests.

### Author Contributions
- Hege Selheim Rinke conceived and designed the experiments, performed the experiments, analyzed the data, contributed reagents/materials/analysis tools, prepared figures and/or tables, authored or reviewed drafts of the paper, approved the final draft.
- Clara Beate Gram Gjesdal conceived and designed the experiments, performed the experiments, analyzed the data, contributed reagents/materials/analysis tools, authored or reviewed drafts of the paper, approved the final draft.
- Heidi Markussen authored or reviewed drafts of the paper, approved the final draft.
- Jörg Assmus analyzed the data, contributed reagents/materials/analysis tools, prepared figures and/or tables, authored or reviewed drafts of the paper, approved the final draft.
- Gerd Karin Natvig conceived and designed the experiments, analyzed the data, contributed reagents/materials/analysis tools, authored or reviewed drafts of the paper, approved the final draft.

### Human Ethics
The following information was supplied relating to ethical approvals (i.e., approving body and any reference numbers):

The study was approved by the Regional Ethics committee for Medical Research (REK, 2011/490).

### Data Availability
The raw measurements are available in the Supplemental File.

### Supplemental Information
Supplemental information for this article can be found online at http://dx.doi.org/10.7717/peerj.6771#supplemental-information.

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
