# Peer review of "Patient-reported fatigue in patients with rheumatoid arthritis who commence biologic therapy: a longitudinal study"

_PeerJ, doi:10.7717/peerj.6771_

## Round 0.1 · original submission · Major Revisions

Thank you for this interesting manuscript on fatigue in RA. The data include only 48 participants and it is unclear how their specific biologic treatment and their level of fatigue relates. It seems quite a leap to assume that all biologics would act in the same manner with all patients in relation to fatigue. I don't think the information you provided in the second paragraph in relevance to clinical practice is relevant here as you did not provide an intervention, in particular a non-pharmacological intervention.

·

Basic reporting

Generally the reporting is clear, though there are some grammatical errors, some examples attached in the abstract, but would suggest a full review prior to resubmission

Experimental design

The aouthers defined the question well, though it was in a very specific population.
Overall the experimental design is fine are seems to have been carried out well with some minor reporting issues. The authors should have considered extending the recruitment period as 48 patients is an extremely small number to make any real assumptions about. Also the authors should have adjusted the analysis using baseline ffs.

Validity of the findings

It is difficult to make relationship assumptions based on such a small sample size. There is a great risk of type II errors and this may lead to an overemphasis on gender fatigue - also not assessed based on baseline FFS. I think the authors need to be clear in the abstract that this is an issue as well as doing adjustment analysis.

Also the authors haven't considered the role of individual biologics in this setting - these biologics have varied mechanisms of actions and this could lead to issues around fatigue. Given that there are five biologics it would be difficult to show statistical differences in the magnitude of effect in these numbers (compared to two genders).

Additional comments

Suggest needs to consider the reporting of the methods - critical appraisal checklist added to attached document

Reviewer 2 ·

Basic reporting

The manuscript is generally well written, but could still be improved, and some sentences need clarification:
In lines 124-125 you write: "In a previous study of both self-reported and physician reported data,…". I assume that it was the outcomes that were studied, not the data, and suggest that you re-write this sentence.
In general, I would prefer the concept patient-reported outcomes (PROMs) rather than self-reported. You will find this in many publications.
Moreover, you use both physician-reported and health personnel-reported data. You refer to DAS28 as “The most common physician-reported measure”. As far as I can see, it was a nurse that collected the clinical data in your study. Maybe you could use one term throughout the manuscript to avoid confusion?
In line 132 you state that “One study found higher prevalence of RA in women than men”. The gender difference in prevalence of RA is well-documented in many studies. I suppose that this is not your point here?
Study design: The setting for the study is not reported. I assume that it was a rheumatology specialist department?
Ethical considerations: How were the patients informed? Oral and/or written information?
Minor comment: The in text citations do not consequently follow the required reference style (3 Authors et al....)

Experimental design

Inclusion criteria: Could the patients consent to participate in the main study and decline participation in your study, or did they consent to participate in both studies at the same time?
Assessments: Does the FSS have a defined cut-off for severe fatigue?
Do you know if the psychometric properties of the instruments have been tested in Norwegian populations?
Lines 196-198: You refer to DAS28 as physician-reported data. To my knowledge DAS28 also includes Patient Global Assessment of Health (PGA), see for example https://www.nras.org.uk/the-das28-score. Did you include that?
Your second aim was to identify possible predictors for change in fatigue. Did you have any hypotheses that guided your selection of possible predictors?
In the abstract (lines 35-36) you state that you performed bivariate and multivariate regression analyses. In line 205 in the Methods section you state that you have used “Linear mixed effect analyses to identify associations between change in fatigue level and clinical variables, such as…” I suggest that you reformulate this sentence to “…analyses were used to identify possible predictors of…” to make clear that these analyses are related to your second aim. Moreover, I suggest that you use predictors when you refer to the results of these analyses in the Discussion.

Validity of the findings

The results are sufficiently reported in tables and text and are consistent with other studies of fatigue in patients with RA.
In line 239, you write that you "did not observe any effect of DAS28 on fatigue". I guess you mean that DAS28 was not a significant predictor of fatigue, i.e. change in fatigue was not explained by disease activity?
You might also have a look at a study by Olsen, C. et al Arthritis Care an Research 2016, which is related study With similar results in Norwegian RA patients.
Discussion, line 268: I do not think that “Patient reported versus reports by health professionals” is a subheading that covers the content written below. Neither is it related to the aims of the study. I suggest deleting this subheading and rather using the two aims as subheadings. Moreover, I miss a description of the strengths of the study.
Line 299: I do not think that the reference to a study of fibromyalgia patients is relevant here as your study focus on patients who commence biologic therapy.
Line 315: The sentence “… as the Norwegian population is rather homogeneous” could be supported by a reference, for example to studies of other RA populations in Norway. I am not sure if the population in general is rather homogeneous.
In the last paragraph, you might as well refer to the study of a group cognitive behavioural therapy program for fatigue by Hewlett et al 2011 and a Norwegian mindfulness-based programme by Zangi et al 2012.

---

## Round 0.2 · Minor Revisions

A majority of the reviewers concerns have been addressed. A few more issues need to be addressed.

·

Basic reporting

none

Experimental design

With regards to #3 The authors should have adjusted the analysis using baseline FSS. You answered that "Thank you for pointing out this. We used a linear mixed effects model including all time points, i.e. also FSS at baseline. In time domain we used a simple contrast, which means comparison of each follow-up time point with baseline. We have added a sentence to clarify this. See page 10, last paragraph."
I am still unsure whether the baseline difference was adjusted for, in general the default (in most software) is to look a them independently form each other, this would mean that the adjustment wouldn't be taken into account. My concern is that there is a statistical difference at baseline between gender, which could be due to other factors. Females have worse fatigue at baseline and treatment would lead to greater improvement in more severe conditions with biologics. Given the small number in each group it wouldn't take much to for this to be overestimated hence the importance of getting the analysis right - is it severity or gender? This would not be taken into account in default analysis some other methods would have to be used (e.g autocorrelation) to include baseline ffs. This may have been carried out but it is not clear from the text further clarity is needed, and if not done correctly should be redone.

Validity of the findings

none

Additional comments

none

Reviewer 2 ·

Basic reporting

My major concerns in this section have been sufficiently addressed. I have only one comment to the inclusion of study section: "The study was carried out at a rheumatology specialist department (not by a department). I would also suggest to include in which part of Norway this was, as you refer to this in the Discussion section.

Experimental design

Re. my comment on hypothesis for selection of possible predictors: I think that the sentence you have added on pg. 7does not provide any further information and may be deleted. I realise that this is an explorative study and that the possible predictors are sufficiently accounted for in the Background.

Validity of the findings

I have no further comments to this section.

---

## Round 0.3 · accepted · Accept

Thank you for your updated submission.